# Using Arts-Based Methodologies to Understand Adolescent and Youth Manifestations, Representations, and Potential Causes of Depression and Anxiety in Low-Income Urban Settings in Peru

**DOI:** 10.3390/ijerph192315517

**Published:** 2022-11-23

**Authors:** Liliana Hidalgo-Padilla, Ana L. Vilela-Estrada, Mauricio Toyama, Sumiko Flores, Daniela Ramirez-Meneses, Mariana Steffen, Paul Heritage, Catherine Fung, Stefan Priebe, Francisco Diez-Canseco

**Affiliations:** 1CRONICAS Center of Excellence in Chronic Diseases, Universidad Peruana Cayetano Heredia, Lima 15074, Peru; 2People’s Palace Projects, School of English and Drama, Queen Mary University of London, London E1 4QA, UK; 3Unit for Social and Community Psychiatry, Queen Mary University of London, London E13 8SP, UK

**Keywords:** depression, anxiety, common mental disorders, adolescents, young adults, youth, arts-based research, Peru

## Abstract

Background: Arts-based methodologies can be beneficial to identify different representations of stigmatized topics such as mental health conditions. This study used a theater-based workshop to describe manifestations, representations, and potential causes of depression and anxiety as perceived by adolescents and young adults. Methods: The theater company *Teatro La Plaza* conducted three online sessions with a group of adolescents and another with a group of young adults from Lima, Peru. The artistic outputs, which included images, similes, monologues, and narrations, were used to describe the experiences of depression and anxiety symptoms following a content analysis using posteriori categories. Results: Seventeen participants joined the sessions. The artistic outputs showed: physical, behavioral, cognitive, and emotional manifestations of depression and anxiety; a perception that both disorders have a cyclical nature; and an awareness that it is often difficult to notice symptom triggers. The mandatory social isolation due to the COVID-19 pandemic was highlighted as an important symptom trigger, mostly linked to anxiety. Conclusions: The findings are consistent with the literature, especially with regard to the manifestations, representations, and potential causes that trigger depression and anxiety. Using arts-based methods allowed adolescents and young adults to expand the articulation of their representations of mental disorders.

## 1. Background

Globally, it is estimated that 1.2 billion people are aged 15–24 years old, representing 16% of the population [1]. One out of seven people (14%) within this age range experiences mental health disorders, of which depression and anxiety disorders are the most common, with a global prevalence of 4.6% and 3.4%, respectively [2]. The COVID-19 pandemic has significantly increased the prevalence of such disorders, with more than 25% of youths showing depressive or anxiety symptoms in some low- and middle-income countries [3,4,5]. This could be explained by the increased risk of abuse or neglect and feelings of vulnerability under unstructured environments among adolescents [6] and by the context exacerbating the stress of entering the adult age [7,8].

Various studies have highlighted the high treatment gap for mental health care among young people [9,10]. For example, in 2020, 83.7% of informants caring for an adolescent in Peru reported not seeking any help for them, although they noticed an emotional problem was present [11]. A study in Brazil, emphasizes the structural, psychosocial, and demographic factors hindering access to treatment [10], while another study in Australia stressed that even though adolescents acknowledge their need to seek help, they did not do so [9]. 

It has been hypothesized that understanding how this population perceives mental health disorders could help develop specific strategies to increase treatment-seeking and adherence [12]. Qualitative studies using interviews and focus groups in the UK [12] and Spain [13] found that adolescents and young adults with a history of anxiety or depression usually reported various symptoms and highlighted the uniqueness of experiences for each person. Although these studies have described how young people perceive and understand depression and anxiety, these descriptions are built from structured discourses of knowledge and experiences.

In contrast, arts-based research methodologies can be beneficial in providing perspectives not typically addressed in research. Arts-based research is a new transdisciplinary paradigm building on the similarities between arts and science [14] using arts as the primary data collection and analysis method throughout research [15,16]. Often employed as an expansion to conventional qualitative methods [17], arts-based research can provide more nuanced perspectives than other research methods [18]. Arts methodologies allow participants “to explore and play with knowing and meaning in ways that are more visceral and interactive than the intellectual and verbal ways” [15], which can promote the expression of their subjectivity more creatively by using other types of language [19]. This is particularly useful when conducting research with children and young people, who may not communicate in the coherent and linear ways usually demanded by qualitative methods such as interviews and focus groups [20,21].

Within health research, arts-based research methodologies have been used to explore health topics that, due to their stigmatized nature, might be difficult to discuss among young people such as body image, reproductive health, depression, and suicide [22,23], some of them using theater as a resource [24]. Previous studies have shown that these methodologies can depict ‘a rich and detailed picture of the illness experience’ [17], particularly because it enables participants to express their emotions [25]. Arts-based research has been successfully used to explore youth lived experiences of anxiety [26] and depression [27], the students’ mental health [28,29] and children’s well-being [30].

The present study used arts-based methodologies to gather the representations of depression and anxiety among adolescents and young adults living in low-income urban communities in Lima, Peru. This study, conducted as part of the project “Building resilience and resources to reduce depression and anxiety in young people from urban neighborhoods in Latin America (OLA)” [31], had two specific objectives:To identify and describe the physical, behavioral, cognitive, and emotional manifestations of depression and anxiety as perceived by adolescents and young adults.To describe the potential causes of depression and anxiety symptoms identified by adolescents and young adults.

## 2. Materials and Methods

### 2.1. Design

We conducted a qualitative study using a phenomenological-hermeneutical approach in order to understand the experiences of the subjects of the study [32]. This design was selected to describe, understand, and reflect on the young people’s experiences and representations of depression and anxiety, which were evoked during dramaturgy-based workshops. The workshops were part of a collaboration between the research teams from Universidad Peruana Cayetano Heredia and Queen Mary University of London, the theater company *Teatro La Plaza* and the research center People’s Palace Projects. They were designed to motivate participants to express themselves about their experiences of depression and anxiety through different creative exercises.

### 2.2. Participants

Participants were former and active students of art programs delivered by *Arena y Esteras*, an arts organization that has collaborated with *Teatro La Plaza* on previous projects. *Arena y Esteras* runs art programs involving theater and circus in low-income urban settings in Lima, Peru’s capital city.

The inclusion criteria comprised (a) being between 15 and 24 years, (b) being willing to take part in arts-based activities, (c) participating or having previously participated in at least one of *Arena y Esteras’* creative programs to guarantee that they are familiar with art activities, (d) having consent provided by a parent/guardian (for participants under 18 years of age), and (e) having the capacity to provide assent (for participants under 18 years of age) or consent (for participants above 18 years of age). Potential participants were excluded if they (a) had a diagnosis of severe mental illness (psychosis, bipolar disorder, schizophrenia), (b) had a diagnosis of cognitive impairment, or (c) were unable to provide informed consent or assent.

### 2.3. Dramaturgy-Based Workshops

*Teatro La Plaza* designed a dramaturgy-based workshop to be delivered online to abide by the social distancing restrictions due to the COVID-19 pandemic. Participants were allocated to two groups according to their age, based on the premise that young people feel more comfortable sharing their experiences with people of a similar age and currently doing similar activities (e.g., school, work). The first group included adolescents between 15 and 17 years old, and the second group included young adults between 18 and 25 years old. The cut-off age also marks the usual end of school years, so the research team assumed that participants in each group will have different daily experiences. Both groups involved male and female participants.

Each workshop consisted of three weekly group sessions of approximately 2 h conducted by one dramaturg and one actor from *Teatro La Plaza*. Each session focused on a specific topic. Session 1 explored the manifestations and representations of depression and anxiety, session 2 explored the potential causes that could trigger these disorders, and session 3 explored the strategies that helped them cope with emotional distress. The final session’s content was analyzed in another publication [33]; thus, it was not included in the present study.

During the workshop, all participants received introductory explanations about dramaturgy and later applied this knowledge to their creation of artistic outputs around anxiety and depression (see Table 1) such as drawings, similes, scripts, and monologues. Some activities were individual, while others were in groups. After each art activity, the research team from the Universidad Peruana Cayetano Heredia conducted a 15 min semi-structured conversation to explore the same topic covered in the activity. In between the group sessions, participants received an individual and a group mentoring session where they received assistance with their arts outputs from one of the facilitators from *Teatro La Plaza*. The outline of the workshops is detailed in Figure 1.

### 2.4. Procedure

A list of potential participants who met the inclusion criteria was filtered by *Teatro La Plaza* and *Arena y Esteras* from all of the young people engaging in their art activities. They invited potential participants to a video call meeting, where the research team explained the study’s objectives and procedures, and *Teatro La Plaza* explained the workshop’s structure and content. 

After the initial meeting, young people who wanted to participate in the study received an informed assent/consent document via WhatsApp. Parents of underage potential participants also received a copy. The following week, two members of *Arena y Esteras* visited their homes to collect the signed documents from the participants and their parents and provided them with the workshop materials.

The online workshops took place over three consecutive weeks in August and September 2020 and were recorded using Zoom VoIP (Voice over Internet Protocol) software v5.9.3 (Zoom Video Communications, San Jose, CA, USA) [34]. In addition to the participants and the *Teatro La Plaza* facilitators, three members of the research team and one member from *Arena y Esteras* were present as observers and/or to address any issues. Additionally, one of the research team members led the semi-structured conversations. The artistic outputs were stored by the research team as images (drawings), videos (performances), or text (similes, scripts) and protected with passwords.

### 2.5. Analytic Strategy

Our analysis was based on a phenomenological-hermeneutical framework for qualitative studies [32]. At the phenomenological level, the artistic outputs from each participant were used to describe the actions, behaviors, intentions, and experiences portrayed in them. At the hermeneutical level, the parts (each of the artistic materials) and the whole (the integration and relationship between the materials produced by each participant) of the artistic outputs were analyzed and focused on specific characteristics through a dialectical process to achieve an adequate comprehension of them.

A summary of the artistic outputs created during the workshops and their association with the research objectives are detailed in Table 1. Artistic outputs were organized into sets per participant, comprising the four individual outputs and the narration that was written in couples. This organization into sets allowed the research team to better understand some of the specific outputs that initially seemed vague or indistinct. For example, the description of an event in the lifeline could provide context for the image of the same participant.

The coding process varied depending on the type of artistic output, as detailed below.

*Similes and monologue scripts.* The coding of these outputs followed two procedures: (a) content analysis of the explicit text [35], and (b) analysis of literary figures and linguistic and communicational resources. The first enabled the coding of text describing symptoms or conceptualizations, whereas the latter required interpretation from the research team to identify the implicit text.*Images.* The coding of the images involved the identification of three main aspects: (a) the main character or element (e.g., a person, a thing, a place); (b) the denotative message, which refers to the description of the characters, the scene depicted and the actions or behaviors; and (c) the connotative message, which refers to the description of the emotions or sensations evoked by the image and its underlying values [36].*Lifelines and narrations.* Since both outputs narrated real-life situations, the explicit content was coded using content analysis.

Two researchers (AV, LH) reviewed two sets of outputs (one from an adolescent and one from a young adult). These were independently identified through open coding posteriori categories, codes, and subcodes, which were then discussed with another researcher (FDC) to refine them.

Once consensus on the posteriori categories, codes, and subcodes was reached, three research team members (AV, DR, SF) used a coding list to independently code a set of outputs from three adolescents and two young adults (30% of the sample). The coding list was obtained using Atlas.ti 7.5.4 software (Scientific Software Development GmbH, Berlin, Germany) [37].

At this stage, the level of agreement among all researchers was calculated using the reliability estimate alpha (α) from Krippendorff [38] and reached an adequate level of 83%. Disagreements were resolved by conducting in-depth discussions between the researchers (AV, LH, SF, DR, MT). Furthermore, the Atlas.ti software was used to merge the categories, codes, and sub-codes based on their similarities. When necessary, the research team also added or modified new categories, codes, and subcodes using axial coding [39]. Later, the artistic outputs of the remaining 12 participants were distributed between the same three team members and they coded the data. Once saturation of the categories, codes, and subcodes had been reached, a member of the research team reviewed the codes again to assess the reliability, which reached between 92% and 94% agreement (see Table 2).

### 2.6. Ethical Considerations

The study protocol was approved by the Institutional Research Ethics Committee of the Universidad Peruana Cayetano Heredia (E021–03–20) and the Queen Mary University of London Ethics of Research Committee (QMERC2020/02). All participants and a parent or guardian of those under 18 years old went through an informed consent or assent process. Confidentiality was ensured by assigning a code to each participant instead of personal data (such as first and last names) across all study materials and artistic outputs. Identifiable information was only available to the Peru research team.

## 3. Results

Seventeen participants were included in the study workshops. Nine adolescents between 15 and 17 years old (median: 17) were included in the first group, and eight young adults between 18 and 25 years old (median: 19.5) were included in the second group of workshops. In the adolescent group, 55% of the sample was female, while only 38% were female in the young adult group. All participants lived in low-income urban communities in Lima and had previous experience practicing one or more arts activities including theater, circus, and dance, among others. All of the artistic outputs developed during the workshops can be found in Appendix A in their original language.

### 3.1. Description of the Workshop Artistic Outputs

The participants wrote 42 similes, 18 were written by adolescents (eight anxiety and 10 depression), and 24 by young adults (16 depression and eight anxiety). These rhetorical resources allowed us to establish analogies about the symptoms of anxiety and depression. Implicit messages related to the physical, cognitive, and emotional manifestations of anxiety and depression (i.e., loneliness, emptiness, rumination of thought) were identified.

*“When I am sad, I feel as if I was the only human on earth”* (Female adolescent’s simile about depression).

*“When I am anxious, I feel like a scrambled Rubik Cube”* (Male young adult’s simile about anxiety).

Participants also wrote 17 monologues, all of them on the same theme as in their similes. We identified two types of “speakers”, referring to a character who speaks in the first person [40] and expresses their feelings, deliberations, reflections, facts, ideas, etc. The first type, the speaker as an individual different from the author, was the most widely used in the monologues. Participants chose to portray disorders (i.e., anxiety, depression) or symptoms (i.e., emptiness, loneliness, confusion, worriedness) as the main characters to express their monologue’s content. In some cases, the main character was talking to the person suffering from the symptoms.

*“Hello, I arrived all the way to you. Your sadness called me, your argument with happiness makes me stronger. But don’t you worry, I am here to give you shelter. Do you know who am I? Yes, I am depression […]”* (Male young adult’s monologue about depression).

The second type, in which the speaker’s voice is merged with the author’s voice, was used by some participants to show their mood, thoughts, emotions, or perceptions about events happening to them.

*“Today, as usual, I arrived at school and, as usual, she started speaking, not as she did before, but with a kind tone and hurtful words… what is leading me to this, that doesn’t cause me pain, but some sort of satisfaction? I don’t know if it is true, but it feels like it is. I don’t have any value”* (Female adolescent’s monologue about depression).

The 18 images presented by the participants included illustrations downloaded from the Internet and the participants’ drawings (see Figure 2a,b). Most of the main characters in these images were people, followed by animals, things, and places. At the denotative level, we identified that participants associated symptoms of depression or anxiety with actions such as lack of appetite, difficulty sleeping, and self-harm, among others. At the connotative level, the images evoked emotions or feelings such as emptiness and loneliness.

Participants worked on 17 lifelines using threads, markers, and pens (see Figure 3). All of them included a central line, representing the course of their lives, which had ups and downs, circles and spirals. The events included in the lifelines will be further described below, in the section on the causes of depressive and anxiety symptoms.

Finally, the participants wrote eight narrations (four by adolescents and four by young adults) about the events identified in the lifelines, detailing actions, people, spaces/contexts, and times of the onset of symptoms in the plot of the narration. These elements made it possible to identify the triggers of the symptoms of anxiety and depression and the underlying and consequent factors to these, in addition to the different types of associated symptoms.


*“And suddenly, [we were] locked up (…) When the quarantine began, it didn’t shock me much. Sometimes we cooked, we played cards. Later, yes… I started to get bored because my family no longer wanted to do activities with me. I wanted to go out, and they wouldn’t let me even go near the door. As I got used to it, not I feel lazy to go out.*


*I take my classes in bed, watch TV in my bed, talk to friends in my bed, have Zoom parties in my bed, listen to music in my bed… The only thing I don’t do [in my bed] is to eat and go to the toilet.”* (Female adolescent’s narrative).

We defined four main categories to analyze the artistic outputs presented above: (a) Manifestations of depression and anxiety; (b) Conceptualization of depression and anxiety; (c) Depression and anxiety symptom triggers; and (d) Factors influencing the continuity or severity of symptoms. Each of these categories will be described in detail below.

### 3.2. Understanding Depression and Anxiety through the Eyes of Adolescents and Young Adults

Two subcategories identified as part of the analysis explain how adolescents and young adults understand and represent depression and anxiety. First, the manifestations of both disorders aimed to explain how depression and anxiety were perceived by young people when experiencing them. Second, using an artistic expression, the participants provided their interpretation of what depression and anxiety were and how it was to live with them.

### 3.3. Manifestations of Depression and Anxiety

The manifestations of depression and anxiety reported by adolescents and young adults have been categorized into physical, behavioral, cognitive, and emotional. However, it should be noted that the participants sometimes expressed different manifestations in the same output. Most of the manifestations were shown in the similes, images, and monologues, but some could also be extracted from the lifelines and narrations.

Cognitive and emotional manifestations of depression and anxiety were the most commonly reported. For example, loneliness, rumination, a feeling of not belonging, and lack of motivation were common manifestations of both disorders among the two age groups. Emptiness and sadness were also commonly reported as depressive symptoms, and a constant state of alert and distress as anxiety symptoms.

Other cognitive and emotional manifestations of depressive and anxiety symptoms that were identified, although less frequently, were confusion, sense of abandonment, anger, emotional pain, and frustration. In addition, feeling worthless or as defeated fighters were also reported by a few youths as a depressive symptom.


*“When I am anxious, I feel like an abandoned soldier”*


*“When I am anxious, I feel like a defeated boxer”* (Male young adults’ similes about depression).

Concerning the physical manifestations, participants commonly reported physical exhaustion as a depression and anxiety symptom as well as crying and headaches, although less frequently. In addition, they mentioned insomnia, sweating, trembling, and tension as common manifestations of anxiety, while difficulty breathing and, sometimes, chest pain, were manifestations of depression.

*“When I am anxious my hands get sweaty, and I can’t help it. I can’t sleep at night and that means the next morning I have dark and deep eye bags”* (Female adolescent’s monologue about anxiety).

Finally, regarding the behavioral manifestations of depression and anxiety, variations in food intake were the most reported, although they were more commonly associated with anxiety. Interestingly, anxiety was mainly linked to increased food intake, whereas depression was related to increased and decreased food intake.

*“My mother complained many times at school [about] the jokes targeting me, but for me, those were not jokes. It was not only the way they talked, but it was also that I am a foreigner. (…) Since that first time, I made a decision, I would mimic a way of talking I didn’t even know, nor was mine. I started practicing at home that way of talking, and I was anxious about learning it, I was eating a lot. Every time I went to school, I didn’t want to speak to anyone, I sweated a lot and my hands were shaking”* (Female adolescent’s narration of a life event included in her lifeline).

Other frequently reported manifestations were self-harm for anxiety in adolescents and depression in one young adult (see Figure 4) as well as behaviors of defiance and opposition to authority figures in the family, school, and neighborhood as a symptom of depression and anxiety among both age groups. Yelling and avoiding social interactions also reflected depression and anxiety, respectively. 

### 3.4. Conceptualization of Common Mental Disorders

To conceptualize depression and anxiety, similes and monologues were used. Within these artistic outputs, one of the most recurrent topics was the length and course of these disorders. For example, some of the participants’ outputs revealed that they did not realize the presence of depressive or anxiety symptoms until after experiencing them for some time, or that there was no way to anticipate when an episode would arise. The outputs also highlighted it was impossible to anticipate how long the symptoms would last, and that these tended to have a cyclical nature.

*“(…) I am here to make you company. As much time as you allow me to.”* (Female young adult in a monologue impersonating depression).

*“Well, it was a pleasure talking to you… I will come back when you least expect it…”* (Male adolescent in a monologue impersonating depression).

Another finding is that depression and anxiety were sometimes described as “friends” that accompanied the young person when coping with difficult situations and it was not until some time had passed that they realized how difficult it could be to get rid of them. The participants’ outputs also expressed a feeling of symptoms merging with their personalities, and that in the long-term, symptoms would become a part of them.

*“I can help you whenever you need it, I can be your friend and partner forever.”* (Female adolescent in a monologue about depression).

*“And if I stopped worrying? No, I can’t. The anxiety is a part of me.”* (Female adolescent in a monologue about anxiety).

### 3.5. Causes of Depression and Anxiety Identified by Adolescents and Young Adults

The two subcategories identified as potential causes were split into (a) triggers that could induce symptoms of depression and anxiety, and (b) factors influencing the continuity and/or severity of symptoms. These subcategories were obtained from the images, lifelines, and narrations.

#### 3.5.1. Triggers of Depression and Anxiety Symptoms

The triggers identified during the analysis were at the individual, family, social, and educational levels.

At the individual level, the most common triggers for both depression and anxiety include having suffered an accident or illness that put someone’s own life at risk or going through medical intervention or surgery, both reported by adolescents and young adults, and not achieving an important personal goal, only mentioned by young adults. Although less frequent in both groups, other triggers were romantic relationship issues or a negative body image.

*“I fractured my finger, this finger, and the surgery was expensive for us at that time (…) we were paying for a motorcycle I used for work and I couldn’t work for almost a month (…) I was feeling depressed, well, anxious. First of all, because of the surgery I was going to go through, I did not really know it was a surgery, how it was, what was going on, or what they put on you. And at that moment I felt… I felt scared about what they were going to do to my finger and all that. But I also [felt] scared when I was there, in bed. I could still move, but I couldn’t work because the wound would open, [which] would be worse (…)”* (Male young adult’s description of a life event while working on his narration).

*“In 2018, we had a tour in Germany. In the second show, I had to use a unicycle (…) At the moment I had to do the pendulum, the chain in the unicycle loosened up a bit and during the fall, I fell with all my weight onto my right wrist (…). Once the show finished, they took me to the doctor, they put on a splint, and I could not move it for three months. I was crying a lot, I felt angry and sad at the same time, like a whirl of emotions, washing away all the emotions because of what happened to my wrist (…)”* (Female young adult’s description of a life event while working on his narration).

Other situations identified as potential triggers for anxiety included the feeling of not being able to make their own decisions about going out with friends or performing certain daily activities due to illness, or fear of performing in public in the young adult group.

At the family and social level, in both groups, it was found that one of the most significant triggers for depression and, to a lesser extent, anxiety symptoms, was the death of a family member or a significant person, closely followed by a relative going through a life-threatening accident or illness (see Figure 5).

Other events identified as anxiety and depression symptom triggers included migration or changing houses and being physically distant from relatives. Situations related to difficulties in the relationship with parents such as constant arguments or distant relationships were also linked to depressive symptoms for both age groups (see Figure 6).

Furthermore, family violence, involvement in youth gangs, disagreements or arguments with friends, and losing their trust in a significant figure were also identified as triggers of depression symptoms in the participants’ outputs.

At the educational level, the adolescents highlighted that changing schools, having exams, failing a course or having difficulty giving presentations at school could trigger anxiety symptoms. Similarly, young adults reported that starting or changing universities, being rejected at admission to the university or suffering bullying were possible depressive symptom triggers, while difficulty choosing a career was a source of anxiety.

*“(…) the day I received the notes, I found that I was missing a single point to be able to enter. I was stunned, afflicted, I did not know what to do.”* (Male young adult narrating a life event associated with depression).

Finally, within the COVID-19 pandemic, participants from both age groups, especially adolescents, highlighted mandatory social isolation as an important symptom trigger, mostly linked to anxiety symptoms.

#### 3.5.2. Factors Influencing the Continuity or Severity of Symptoms

Only a few adolescents mentioned issues that could affect the severity or continuity of depression and anxiety including changes in the family structure after a loss, the prolonged absence of family members, experiencing bullying and xenophobia.

*“It was just any morning, I was in fifth grade (…) I arrived home and found my aunt, [who] started saying stuff like “your father was a good man”, “your father will go to one place and won’t come back”, “he went to a good place”. (…) When I found out, I was devastated and, on the next day, I went to my father’s house for his wake and to see him for the last time. Without him, I felt that I was really lonely and sad, I needed my family, but everything worsened when I felt they were distancing me. My brother moved out with his father, and my mother worked too much and didn’t pay attention to me (…). And I felt lonely, I started being a little aggressive towards my family and it was because they were not there for me when I felt bad; I cried a lot when I was alone and I didn’t speak to anyone (…).”* (Male adolescent’s description of a life event while working on his narration).

Young adults, especially males, mentioned that constantly self-evaluating their work, study, or family decisions could increase their symptoms of depression or anxiety.

*“(…) You feel anguish… thinking that you have to choose between limited options and (…) then not being clear about what you want to do or thinking about (…) not achieving what you wanted and thinking that you did not put effort into it (…) I also realized that sometimes there was no support (…) so much so that I felt anguished, I felt anguished because I didn’t know what to do.”* (Male young adult description of a life event).

Other factors identified included overprotection from parents; other people’s negative perceptions about their capacities or abilities; transitioning from school to graduate studies; difficulties adapting to new settings; and pursuing ideal models based on social media characters (e.g., Instagram, TikTok, Facebook) that encourage comparisons between the life models shown and the one they experienced.

*“By that time, I had become a little sedentary, I stopped practicing dancing because I felt tired. I thought “well, we are in quarantine, I will rest, I’ve been dancing all the summer”. Whenever I had free time, I would start watching videos of other dancers, I used more social media and, little by little, I was comparing myself to them. I was telling myself: “wow, they dance so good, I’d love to dance like that. I want that way of dancing”. So I started feeling frustrated, being in a bad humor, to grump, even feeling desperation because I wasn’t doing the dance moves as I wanted.”* (Female young adult description of a life event).

Finally, in both age groups, economic deprivation was a constant stressor that sometimes hindered addressing their problems.

*“Anguish is when I, well, my family decided to move to Lima. (…) but when I got here I realized that we had no money, no money, no place to stay, I felt stressed, I really wanted to go back, to go back to where we lived before.”* (Male young adult description of a life event).

## 4. Discussion

Our study used arts-based methods to understand the representations of depression and anxiety in Peruvian adolescents and young adults. The results were based on the analysis of five different types of artistic outputs and organized into four categories: (a) Manifestations of depression and anxiety; (b) Conceptualization of common mental disorders; (c) Depression and anxiety symptom triggers; and (d) Factors influencing the continuity or severity of symptoms.

Some of our main findings are consistent with those reported in previous research. Undoubtedly, cognitive and emotional manifestations were perceived as core elements of depression among our participants, as has also been found by adolescents and young adults in countries such as Brazil and the United States [41,42]. Some common feelings and thoughts described by adolescents and young adults about depression include sadness, frustration, isolation, despair, loss of pleasure, and hopelessness [41,43], which were also found in our study. Furthermore, adolescents and young adults in our study identified a constant state of alertness and feelings of nervousness as predominant when experiencing anxiety. Likewise, young adults in Portugal [44] and adolescents in Spain [45] reported that different situations or stimuli could be perceived as threatening when experiencing anxiety, and that this would increase their feelings of mistrust in themselves and their environment [44,45,46].

Regarding the physical and behavioral manifestations of depression and anxiety, difficulty breathing, sweating and shaking have also been found in previous research among youth [12,47]. In addition, a lack of social interaction, isolation, self-injurious behaviors and a change in eating patterns were also mentioned as associated with depression and anxiety [12,41,47]. Our findings showed no differences between the manifestations of anxiety and depression among age groups. An idea that did not arise in our findings was the role of the family and peers on their experience of depression or anxiety. As a previous arts-based study showed, relationships or interactions with significant people in their life were often mentioned as linked to the experiences of depression [27].

Concerning the conceptualization of the disorders, the participants highlighted their cyclical nature, which is consistent with the literature about relapses [47,48], but also expanded on how the symptoms took them by surprise or that it was difficult for them to anticipate the consequences, showing a lack of control. In a way, they welcomed the symptoms or the condition before realizing that it would be difficult to recover from them. The constant presence of symptoms can lead to debilitation in young people and the feeling that the symptoms are too strong to be prevented [12,42]. Interestingly, some participants referenced the figure of defeated fighters when referring to anxiety, which gives the impression of unsuccessfully fighting the emotion, which is not far from the figure of anxiety as a monster found in another arts-based study [26].

Two other representations of the disorders included seeing depression as a “friend” that accompanies the young person and as a characteristic that becomes part of their personality, which coincides with a stage of developing identity. Although we did not find any literature related to the first, two articles supported the latter, stating that the symptoms become part of the youth’s sense of self [12], and the need to hide that part of their self to face the realities of their social world [23], as if they had the expectation of seeing it as an external feature, rather than internal. Interestingly, another study exploring depression through theater found that depression was represented as having a mask covering their symptoms and experiences [23]. This shows that using the arts can expand the knowledge of the experiences of distress in youth for researchers, which could also help mental health professionals understand young people’s experiences of distress, and, in turn, improve the rapport between the patients and staff.

In our study, most of the attributed causes and aggravating factors of depression and anxiety included daily life worries such as relationships with family and friends, making life decisions, work or study-related concerns, and economic constraints as well as isolated events such as illnesses, deaths, and changing houses. Even though several studies have also found these causes [12,13,43,49,50], young people also highlight the importance of biological predisposition, self-concept and the interpretation of personal experiences [12,13,49]. It could be possible that due to the workshop design, young people felt more comfortable mentioning external events as potential triggers rather than internal characteristics or that the workshop design did not allow follow-up questions on the artistic outputs to have more depth. Another interesting result is that while potential triggers of both disorders might differ between adolescents and young adults, the manifestations of symptoms and their conceptualization were similar across both age groups. Surprisingly, there were no differences across types of conditions and gender.

Particularly within the COVID-19 pandemic, in which Peru was one of the most affected countries worldwide [51], adolescents and young adults participating in the workshops stressed how social isolation and, thus, the lack of social connection put them at risk of developing depressive or anxiety symptoms. This has also been raised by a systematic review of the mental health experiences of adolescents globally [52]. Additionally, an international study also found that a lack of social support, among other variables, consistently predicted poorer mental health outcomes throughout the pandemic [53].

Furthermore, we emphasize the importance of using arts-based methods as a tool to collect sensitive data from adolescents and young adults. As described in previous studies [54,55], collecting sensitive data with some populations such as adolescents can be challenging because it requires a long time to deepen into representations that do not arise easily using logic; therefore, the use of arts can be an alternative route to access a wide array of thoughts and experiences. This method also led us to innovative findings on the understanding of depression and anxiety representations, which we did not find in the available literature. Interestingly, other qualitative studies exploring similar topics included quotes in which youths used similes like the ones we proposed as part of our activities [27]. Although they did not intend to use arts purposely, the arts language seems to be familiar to young people when expressing thoughts and opinions.

*One participant said his depression: “Is like a water jar with a leak…It just empties little by little. It fades, but not at once. But then it is also like someone grabs it and turns it over because they can’t stand this slow leak […] So it’s just empty and they can start over, buy a new one.”* (Boy 1, 15) [42].

The artistic language adds nuance to the sensations and emotions produced by the experiences of disorders such as depression and anxiety. Using metaphors to describe them added color and transmitted feelings that may not be described in full detail by conventional words. For example, when one of the female adolescent participants referred to anxiety as a needle picking her skin, she expressed how anxiety represented a corporal experience of pain, but also that she faced a state of alert. The artistic language merges with the narratives of daily activities experienced by adolescents and young adults, which enables us to understand how the local context shapes the experiences of depression and anxiety. Using characters to embody depression and anxiety enabled us to understand how the symptoms of these disorders accompany the young people living with them.

Using arts-based research, specifically performance approaches, is also innovative because it challenges the idea that data are always objective. As Chamberlain et al. [56] described, arts-based research in psychology allows flexibility for the reader to interpret the research findings largely. Our data analysis involved frequent discussions among the research team in an effort to interpret the data while taking into consideration socially constructed knowledge and contextual connections. Furthermore, the fact that the participants easily engaged with the activities and provided artistic outputs for all of the tasks shows that they felt comfortable with the data collection method. However, as arts-based research is still scarce, there is a need to better understand when this methodology is most helpful to identify new data compared to other tools and to standardize the research procedures to enhance rigor.

### Limitations

The workshop was initially designed for in-person delivery. However, due to the social distancing restrictions as a result of the pandemic, the research team and arts organization decided to shift toward an online delivery. Since the sessions took place shortly after the pandemic began, facilitators and participants were still adapting to remote activities, which might have produced discomfort among participants who were not used to virtual platforms. Another study limitation is that the participants were already engaged in art activities prior to the study, which could be an indicator of accessing resources not all adolescents and young adults have access to; thus, it is impossible to generalize the results for all people within this age group. A final limitation was that the workshop design did not include follow-up questions to the artistic outputs, which sometimes made their interpretation difficult.

## 5. Conclusions

This study allowed us to explore how a group of low-income adolescents and young adults represented depression and anxiety using arts-based methods. A dramaturgy-based workshop enabled the participants to share their views on the manifestations and causes of depression and anxiety. Although some of the findings were similar to those produced in previous research, this method expanded on the knowledge of how young people perceived the process of depression and anxiety and how difficult it was for them to notice when these conditions started and settled. The artistic tools offered an insightful view and we encourage the incorporation of arts-based methods to allow young people to creatively express their views about mental health.

## Figures and Tables

**Figure 1 ijerph-19-15517-f001:**
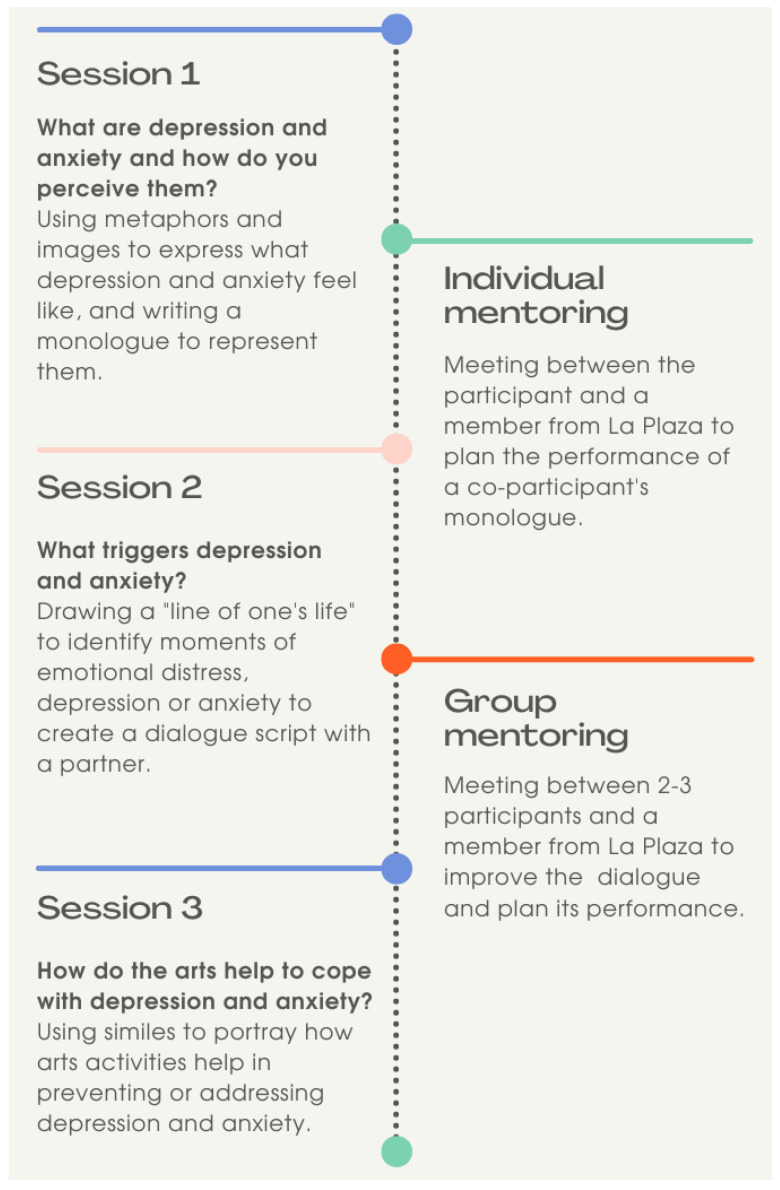
Outline of the dramaturgy-based workshops.

**Figure 2 ijerph-19-15517-f002:**
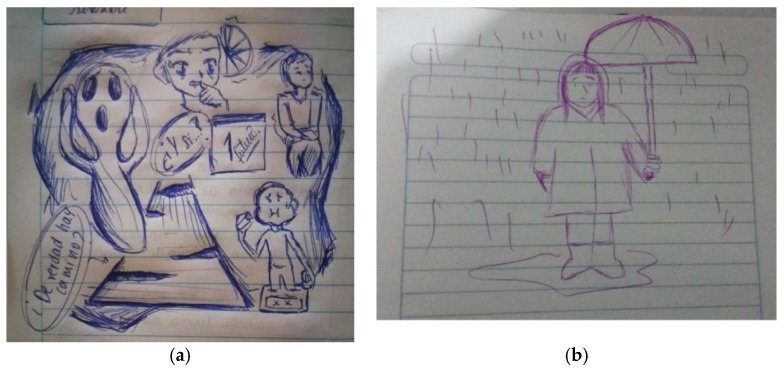
(**a**,**b**) Images of anxiety and depression drawn by an adolescent and a young adult.

**Figure 3 ijerph-19-15517-f003:**
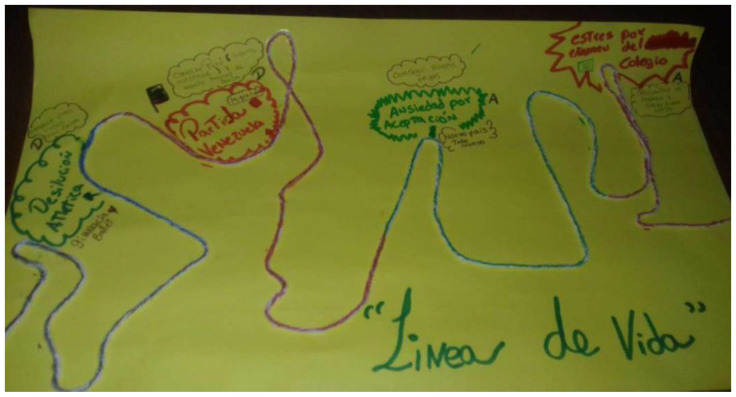
Lifeline drawn by a male adolescent.

**Figure 4 ijerph-19-15517-f004:**
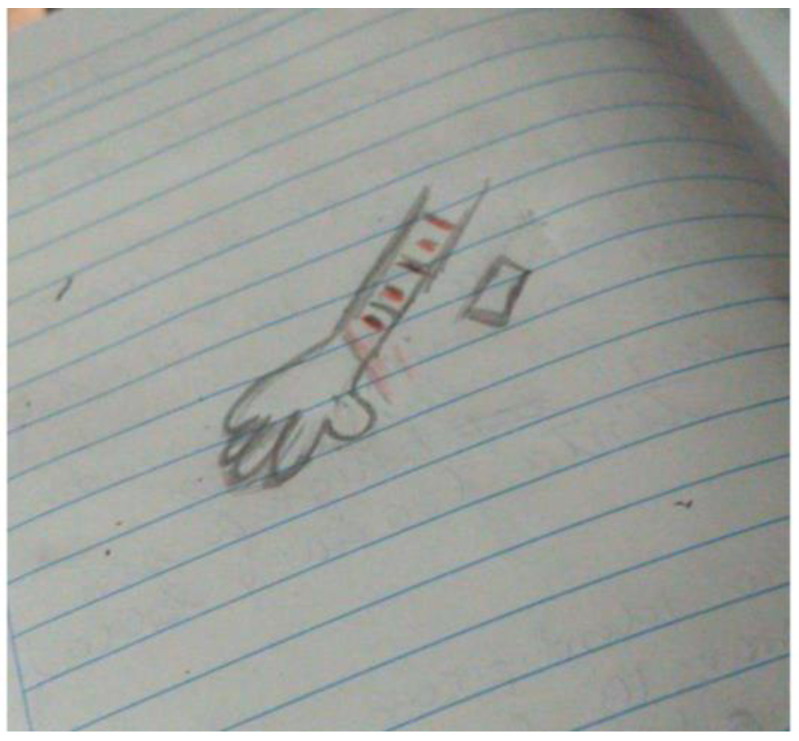
Drawing of self-harm as a manifestation of depression by a young adult.

**Figure 5 ijerph-19-15517-f005:**
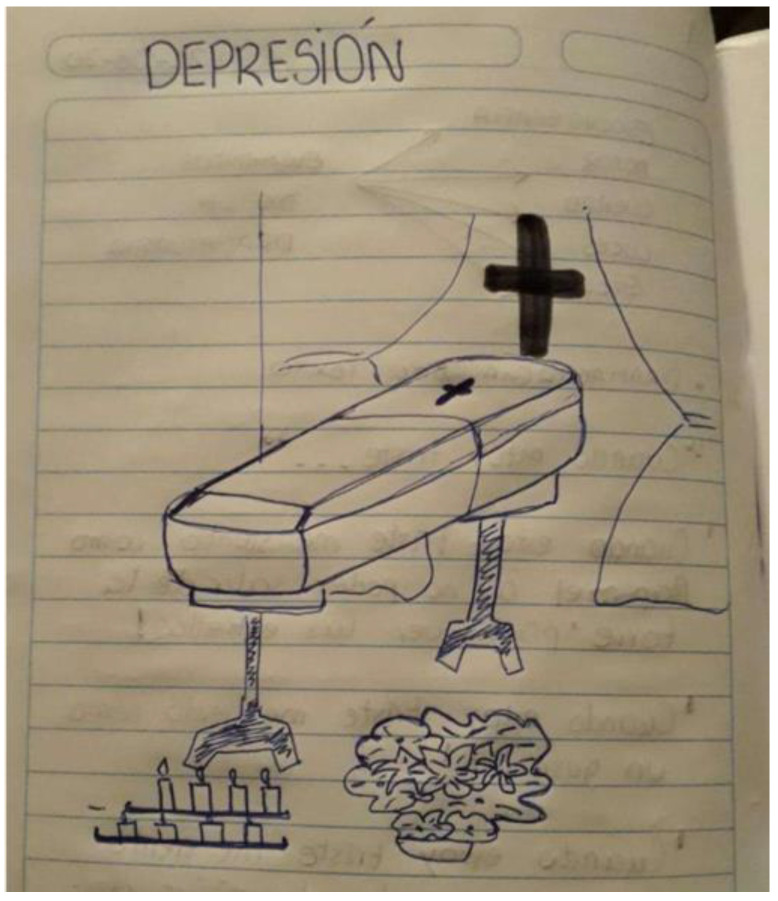
Drawing of death as a representation of depression by a young adult.

**Figure 6 ijerph-19-15517-f006:**
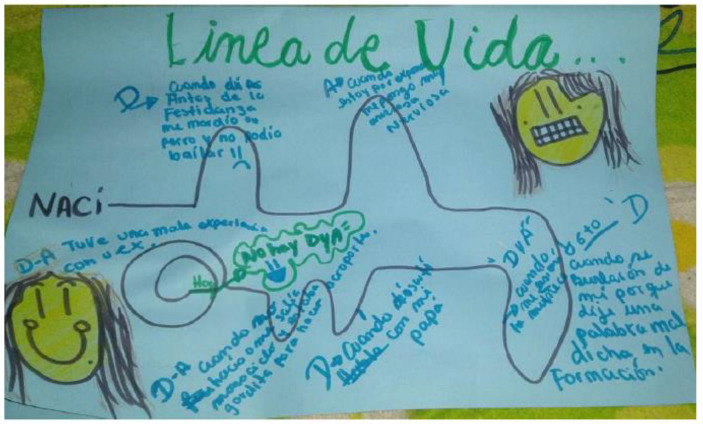
Lifeline of a female young adult including events that triggered depression or anxiety.

**Table 1 ijerph-19-15517-t001:** Artistic outputs from the workshops.

Session	Activity/Output	Associated Objectives	Prompt
1	Similes	Manifestations of depression and anxiety	Write as many phrases as possible starting with either “when I am sad, I feel like…” or “when I am anxious, I feel like…” Example: When I am sad, I feel like a tree that lost all its leaves.
1	Images	Manifestations of depression and anxiety	Choose an image or a picture or draw something representing depression or anxiety.
1	Monologue scripts	Manifestations of depression and anxiety	Imagine depression or anxiety as a character and write a one-page monologue of this character introducing him/herself.
2	Lifelines	Potential causes of depression and anxiety symptoms	Draw a line on cardboard with any shape you would like. This line will represent your life until now. Then, think of four events where you experienced depression or anxiety and write them down on your lifeline. Later, next to each event, write what helped you feel better.
2	Narrations	Potential causes of depression and anxiety symptoms	Discuss the events in your lifeline with a partner. Choose together an event and write a script narrating the event and the strategy(ies) used to overcome the situation.

**Table 2 ijerph-19-15517-t002:** Percentage agreement between coders and the percentage of passages in which the coding categories appeared.

Categories	Codes	Sub-Codes	Frequency (%)	Level of Agreement
Understanding depression and anxiety through the eyes of adolescents and young adults	Manifestations of depression and anxiety	Physical	51 (14.2)	92%
		Behavioral	38 (10.6)	91.7%
		Cognitive and emotional	85 (23.7)	93.3%
	Conceptualization of common mental disorders	Not applicable	17 (4.7)	91%
Causes of depression and anxiety identified by adolescents and young adults	Triggers of depression and anxiety symptoms	Individual level	45 (12.5)	92.8%
		Family and social levels	43 (12)	94%
		Educational level	29 (8.1)	94%
		COVID-19	15 (4.2)	92%
	Factors influencing the continuity or severity of symptoms	Not applicable	36 (10)	93.5%

## Data Availability

The data presented in this study are available on request from the corresponding author.

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
