# Peer review of "Using Arts-Based Methodologies to Understand Adolescent and Youth Manifestations, Representations, and Potential Causes of Depression and Anxiety in Low-Income Urban Settings in Peru"

_ijerph, 2022, doi:10.3390/ijerph192315517_

Round 1

Reviewer 1 Report

General comments

=============

This paper discussed uses a theatre-based workshop to describe manifestations, representations and potential causes of depression and anxiety as perceived by adolescents and young adults, it provides evidence and support on arts-based methods allowed adolescents and young adults to expand the articulation of their representations of mental disorders. I would like to express several concerns and provide some comments and suggestions as follows. Hopefully, the following comments and suggestions will be helpful for improving this paper.

=============

Major comments

---------------------

1. This paper still lacks further explanation of arts-based research methodologies can be beneficial in providing perspectives not typically addressed in research, especially the reference to the past research about the arts-based research and emotion behavior. And why the COVID-19 pandemic has significantly increased the prevalence of such disorders or anxiety symptoms?

2. For the literature, it is recommended that the literature related to compare studies from arts-based research and emotion behavior, in this study, and to indicate the results of studies in other research.  

Minor comments

---------------------

3. This study used the experimental method. And Participants 106 were allocated to two groups according to their age, based on the premise that young people feel more comfortable sharing their experiences with people of a similar age. The first group included adolescents between 15 and 17 years old, and the second group included young adults between 18 and 25 years old. However, their age almost the young age. Do the other control variables influence the result? Ex: education or income?

 4.The authors can discuss results, implications for managers and scope for future researches in different sections to enhance readability

Reviewer 2 Report

This paper fits a particular research line (so-called phenomenological-hermeneutical approach) within the framework of qualitative research. The background of the research is clearly stated. So it is the stage of data collection and its rationale. However, the Authors could improve reader's understanding of the later stages of content units extraction, coding and analysis. Krippendorff's alpha is an adequate divergence measure for evaluation of unstructured material. But Authors should make explicit the additional criteria followed (i) to settle the disagreement, (ii) to specify denotative from connotative content, (iii) to avoid that different alleged units are assigned to the same coding label across so different kinds of subjects' outputs. In this connection, the content analysis software provides useful tools to calculate the 'groundedness' of a code label and to visualize the aggregation of data with different media. This improvement may affect the evaluation of the significance of the research content.

As far as Authors' claim is extended to the advantage of art-based methodology (sections 4 and 5), more caution should be exercised. A research work should have been conducted with different comparable qualitative methodologies. Alternatively, some basic descriptive statistics can be used to compare the significant correlations between units and factors found by employing this methodology and those found in the literature. Indeed, the Authors hint at this point but only for the extraction of a single unit (that is, depression as a 'friend'). Another issue to be discussed is the different characteristics of the artistic media involved.

The reviewer understands that following the suggestions about this latter question might require a substantial work. Therefore, the main recommendation is that at least Authors present clearly their assumptions on such issues. to enhance the scientific soundness of the work.
